# Novel Insights of Lymphomagenesis of *Helicobacter pylori*-Dependent Gastric Mucosa-Associated Lymphoid Tissue Lymphoma

**DOI:** 10.3390/cancers11040547

**Published:** 2019-04-17

**Authors:** Sung-Hsin Kuo, Ming-Shiang Wu, Kun-Huei Yeh, Chung-Wu Lin, Ping-Ning Hsu, Li-Tzong Chen, Ann-Lii Cheng

**Affiliations:** 1Department of Oncology, National Taiwan University Hospital and National Taiwan University College of Medicine, Taipei 100, Taiwan; shkuo101@ntu.edu.tw (S.-H.K.); khyeh@ntu.edu.tw (K.-H.Y.); 2Cancer Research Center, National Taiwan University College of Medicine, Taipei 100, Taiwan; 3Graduate Institute of Oncology, National Taiwan University College of Medicine, Taipei 100, Taiwan; 4National Taiwan University Cancer Center, National Taiwan University College of Medicine, Taipei 106, Taiwan; 5Department of Internal Medicine, National Taiwan University Hospital and National Taiwan University College of Medicine, Taipei 100, Taiwan; mingshiang@ntu.edu.tw (M.-S.W.); phsu8635@ntu.edu.tw (P.-N.H.); 6Department of Pathology, National Taiwan University Hospital and National Taiwan University College of Medicine, Taipei 100, Taiwan; chungwulin@yahoo.com; 7National Institute of Cancer Research, National Health Research Institutes, Tainan 704, Taiwan; leochen@nhri.org.tw; 8Department of Internal Medicine, Kaohsiung Medical University Hospital, Kaohsiung Medical University, Kaohsiung 807, Taiwan; 9Department of Internal Medicine, National Cheng-Kung University Hospital, Tainan 704, Taiwan

**Keywords:** MALT lymphoma, Helicobater pylori, T-cell, Tregs, CagA

## Abstract

Gastric mucosa-associated lymphoid tissue (MALT) lymphoma is the most common subtype of gastric lymphoma. Most gastric MALT lymphomas are characterized by their association with the *Helicobacter pylori* (HP) infection and are cured by first-line HP eradication therapy (HPE). Several studies have been conducted to investigate why most gastric MALT lymphomas remain localized, are dependent on HP infection, and show HP-specific intratumoral T-cells (e.g., CD40-mediated signaling, T-helper-2 (Th2)-type cytokines, chemokines, costimulatory molecules, and FOXP3+ regulatory T-cells) and their communication with B-cells. Furthermore, the reason why the antigen stimuli of these intratumoral T-cells with tonic B-cell receptor signaling promote lymphomagenesis of gastric MALT lymphoma has also been investigated. In addition to the aforementioned mechanisms, it has been demonstrated that the translocated HP cytotoxin-associated gene A (CagA) can promote B-cell proliferation through the activation of Src homology-2 domain-containing phosphatase (SHP-2) phosphorylation-dependent signaling, extracellular-signal-regulated kinase (ERK), p38 mitogen-activated protein kinase (MAPK), B-cell lymphoma (Bcl)-2, and Bcl-xL. Furthermore, the expression of CagA and these CagA-signaling molecules is closely associated with the HP-dependence of gastric MALT lymphomas (completely respond to first-line HPE). In this article, we summarize evidence of the classical theory of HP-reactive T-cells and the new paradigm of direct interaction between HP and B-cells that contributes to the HP-dependent lymphomagenesis of gastric MALT lymphomas. Although the role of first-line HPE in the treatment of HP-negative gastric MALT lymphoma remains uncertain, several case series suggest that a proportion of HP-negative gastric MALT lymphomas remains antibiotic-responsive and is cured by HPE. Considering the complicated interaction between microbiomes and the genome/epigenome, further studies on the precise mechanisms of HP- and other bacteria-directed lymphomagenesis in antibiotic-responsive gastric MALT lymphomas are warranted.

## 1. Introduction

The gastric marginal zone B-cell lymphoma of the mucosa-associated lymphoid tissue (MALT) type (gastric MALT lymphoma) represents the most common extranodal non-Hodgkin’s lymphoma [1,2,3,4,5]. The histological manifestations of MALT lymphoma cells of the stomach are characterized by polymorphous mixtures of centrocyte-like cells, small lymphocytes, and plasma cells, but the absence of high-grade lymphoma cells, such as a confluent cluster or sheets of large cells resembling centroblasts or lymphoblasts [1,2,3]. Morphologically, MALT lymphoma cells often invade and destroy the neighboring gastric epithelium, and cause lymphoepithelial lesions [1,2,3]. Tumor cells of gastric MALT lymphoma are located in the marginal zone and are surrounded by the reactive germinal center (GC), and these lymphoma cells are thought to arise from post-GC memory/marginal zone B cells [6,7]. During the B cell maturation process, the primary B cell repertoire generated in the bone marrow can differentiate to immature B cells (expression of a complete immunoglobulin (Ig)M-molecule) through multiple steps of rearrangement of Ig heavy chain and light chain genes [6,7,8,9]. Furthermroe, the immature B cells can transform into mature B cells (express both IgG and IgM) through the alternative splicing of Ig heavy-chain mRNA [6,7,8,9]. These naïve mature B cells can migrate to the peripheral lymphoid organs or extranodal Peyer’s patches-like lymphoid follicles (MALT) and develop into short-lived plasma cells, or enter into the GC to transform centroblasts, following antigen stimulation [6,7]. In the GC, centroblasts can develop into centrocytes through somatic hypermutation and heavy chain class-switching recombination. Furthermore, selected GC B cells may differentiate into post-GC B cells, including plasma cells or memory/marginal zone B cells [6,7]. These neoplastic marginal zone memory B cells may resemble the centrocytes of GC, and share the common immunophenotype markers of normal marginal zone B cells, including the pan-B antigen (CD19, CD20, CD22, and CD79), but they do not express GC B cell profiles, including CD10 and B cell lymphoma (Bcl)-6 [1,2,5,7]. Neoplastic cells of gastric MALT lymphoma cells are positive for surface Igs (most IgM, and less frequently IgG and IgA), CD43, Bcl-2, and complement receptors (CD21, CD35) [1,2,5,7,10], whereas CD5, CD23, and cyclin D1 immunostains are generally negative in MALT lymphoma cells [1,2,5,7,10], although rare cases of CD5-positive MALT lymphomas have been reported [11].

Epidemiologically, gastric MALT lymphoma is characterized by its close association with the *Helicobacter pylori* (HP) infection [3,12]. Wotherspoon et al. first described that HP-related gastritis and the subsequently developed MALT are more frequent in patients with gastric MALT lymphoma [13]. Furthermore, Wotherspoon et al. demonstrated that the eradication of HP using antibiotic treatments resulted in complete remission (CR) in five out of six cases of gastric MALT lymphoma, an imperative result leading to a new era of using first-line HP eradication therapy (HPE) in the management of gastric MALT lymphoma [14]. A systemic review from Zullo et al., including 32 studies with 1408 patients presenting with localized (stage IE and IIE1) HP-positive gastric MALT lymphoma, reported that the first-line of HPE with 7 to 14 days of triple therapy (i.e., a proton-pump inhibitor (PPI) plus clarithromycin, amoxicillin, metronidazole, or other antibiotics) or high-dose dural therapy achieved CR rates of 77.5% (95% confidence interval: 75.4% to 79.7%) [15,16]. In their reviews, Zullo et al. found that the CR rate was high in tumors confined to mucosa or submucosa and located in the distal stomach (antrum, pylorus, and lower body) [16]. As reported by other investigators, we too observed that the distal lesion sites and tumors limited to mucosa/submucosa were closely associated with the CR rate of tumors (HP dependence of gastric MALT lymphoma) [17,18]. Previous studies have shown that the colonization by HP organisms and acquired MALT were most commonly localized in the antrum and lower body of the stomach [19,20]. All this evidence indicates that HP infection plays an essential role in the lymphomagenesis of HP-positive gastric MALT lymphoma.

Although the role of first-line antibiotics in the treatment of HP-negative gastric MALT lymphomas remains unclear, previous reports have revealed that certain patients with HP-negative gastric MALT lymphomas can respond to a first-line antibiotic treatment [21,22,23]. We recently reported that 8 out of 25 patients (32%) with stage IE/IIE1 HP-negative gastric MALT lymphoma achieved CR after first-line HPE therapy, where the diagnosis of a HP-negative status was based on the absence of a histology, rapid urease test, ^13^C urea breath test, and serology [24]. In addition to the case results in our study, we reviewed 22 previously published results of HP-negative gastric MALT lymphoma patients from 1999 through 2016 and showed that first-line antibiotic treatment resulted in a CR rate of 27.9% (68/244) [24]. These findings indicated that bacteria, other than HP, are associated with the development of gastric MALT lymphoma in humans.

Previous studies have demonstrated the mechanisms that link HP-regulated intratumor T cells, HP-triggering cytokines and chemokines, and HP antigen stimuli with B-lymphoid neoplasms of gastric MALT lymphoma [1,4,25,26]. In addition to the classical aforementioned concept, other investigators, as well as ourselves, have demonstrated that HP cytotoxin-associated gene A (CagA) can promote B-cell proliferation through the CagA-mediated activation of phospho-Src homology-2 domain-containing phosphatase (SHP-2) and the subsequent signaling molecules [17,27,28]. In this article, we have summarized the mechanisms of involvement of T-cell-derived signals and CagA-triggering signals in the HP-dependent lymphomagenesis of HP-positive gastric MALT lymphoma. Additionally, we have described whether genetic polymorphism and HP-associated epigenetic changes are involved in the lymphomagenesis of gastric MALT lymphoma. Considering the subsets of HP-negative gastric MALT lymphoma patients that can be cured by a first-line antibiotic treatment, the possible mechanisms of microbiome-associated antibiotic-responsive HP-negative lymphoma were explored.

The literature survey approaches described in this review article use a combination of the publicly available databases PubMed, Ovid, SCOPUS, and Web of Science. In the literature survey, we have used the following key terms: “B cell lineage”, “histological manifestation”, “Immunophenotype”, “T cells”, “cytokines”, “chemokine”, “chemokine receptors”, “regulatory T cells (FOXP3)”, “epigenetic”, “genetic”, “DNA methylation”, “cytotoxin-associated genes”, “natural killer cell”, “macrophage”, “innate lymphoid cells”, “mucosa-associated invariant T”, “microbiomes”, and “microbiota”, in combination with “*Helicobacter pylori*”, “mucosa-associated lymphoid tissue lymphoma”, “gastric lymphoma”, and “gastrointestinal lymphoma”. In addition, we have used “*Helicobacter helimannii*” and “*Helicobacter felis*”, in combination with “mucosa-associated lymphoid tissue lymphoma” and “gastric lymphoma”. However, in this review we excluded “microRNA”, “antibiotics-unresponsive mechanisms-associated molecules, such as API2-MALT1, BCL10, and NF-κB”, genetic changes and chromosomes that contribute to HP-independence, advanced stages, and the progression of gastric MALT lymphoma, for example t(11;18)(q21;q21), t(1;14)(p22;q32), and t(14;18)(q32;q21). The results of the selected articles and our previously published reports were included in this study for reviewing the precise mechanisms of the HP- and other bacteria-related, gastric microenvironment-related, epigenetic-related, or genetic-related, CagA-direct signaling-related pathogenesis of antibiotics-responsive gastric MALT lymphoma. The different mechanisms contributing to the lymphomagenesis of antibiotics-responsive gastric MALT lymphoma are described under their specific subheadings.

## 2. Indirect Interaction Between Tumor-Infiltrating T Cells and B-Cell lymphoma of Gastric MALT Lymphoma

In the early process of the lymphomagenesis of gastric MALT lymphoma, HP-reactive tumor-infiltrating T cells, induced by the HP antigens, can promote the growth and differentiation of B lymphoma cells through CD40-mediated signaling and T helper-2 (Th2)-type cytokines (interleukin (IL)-4, IL-5, and IL-10) [29,30,31,32]. In the murine model of *Helicobacter felis (H. felis)*-induced gastric MALT lymphoma, tumor-infiltrating CD4+ T cells co-express surface markers CD28 and CD69, and produce large quantities of IL-4, but not of interferon-gamma (INF-γ) [33]. In addition, tumor-infiltrating T cells in gastric MALT lymphoma have been found to malfunction in both perforin-mediated cytotoxicity and Fas-Fas ligand-mediated apoptosis that consequently helps in sustaining the growth of these tumors [34,35]. 

The effective communication between two subsets of co-stimulatory molecules of neoplastic B cells, CD80 (B7.1) and CD86 (B7.2), and the CD28 or cytotoxic T-lymphocyte–associated antigen 4 (CTLA-4) of T cells, is thought to play important roles in the tumorigenesis of B cell neoplasms [36,37,38]. Contrary to CD86 that stimulates the proliferation of B cells and the production of IgG by B cells in B cell lymphoma, CD80 downregulates the B cell response and secretion of IgG in B cell lymphoma [39]. de Jong et al. showed that the CD86 expression was found to be significantly associated with the sensitivity of gastric MALT lymphoma (HP-dependent) to HPE, whereas CD80, CD40, and their corresponding ligands were not [40].

In addition to the stimulatory effects of infiltrating T cells on lymphomagenesis, the development of MALT lymphoma is also triggered by self-antigen-stimulating B cell receptor (BCR)-signaling. This was evident from previous findings that lymphoma B-cells of gastric MALT lymphoma express most surface IgM and IgG, as well as occasional IgA [41]. Hussell et al. showed that B-cells from gastric MALT lymphoma can differentiate into Ig-producing cells, and that the cross-linking of mitogen with tumor Ig-producing cells may be involved in the lymphomagenesis of these tumors [42]. Craig et al. reported that human MALT lymphoma Ig antibodies reacted with various self- and foreign-antigens, including *H. sonicate*, IgG, DNA, and stomach extract [43]. Furthermore, Craig et al showed evidences of somatic hypermutation, variable region heavy chain (VH) gene segment (IgHV1-69), intraclonal variation, and a higher complementarity determining regions 3 (CDR3) length of the heavy chains in Ig of human gastric MALT lymphoma using sequence analysis [43]. IgHV1-69 rearrangements of Ig have been reported for MALT lymphomas of the ocular adnexa, parotid gland, stomach, and lung [44,45,46]. A previous study showed that IgVH1-69 functionally acts as an autoimmune gammopathy that is reactive toward human IgG in Waldenstrom macroglobulinemia [47]. Considering the molecular characteristics of polyreactive antibodies, including the IgM class, CDR3 length, and preferential VH gene variation that are detected in gastric MALT lymphoma [48], the autoantigen stimulation with BCRs and its intracellular signaling functionally promote the proliferation and development of gastric MALT lymphoma.

Taken together, these findings indicate that the growth of malignant B cell clones of gastric MALT lymphoma may be dependent on the help of T cells and certain antigens, and suggest that most gastric MALT lymphomas remain localized and regress after first-line HPE, which eradicates the HP-specific T cells and antigens (Table 1).

## 3. Chemokines and Their Receptors, and Regulatory T-Cells in Gastric MALT Lymphoma

Previous studies have demonstrated that chemokines and chemokine receptors are involved in the lymphomagenesis of gastric MALT lymphoma [49,50,51,52,53,54,55]. Mazzucchelli reported that B cell–attracting chemokine-1 (BCA-1) and its chemokine receptor, C-X-C chemokine receptor 5 (CXCR5), are both highly expressed in MALT and lymphoma B cells of HP-positive gastric MALT lymphoma [49], indicating that BCA-1, induced by chronic HP infection-stimulating dendritic cells and B-lymphocytes, is involved in the lymphomagenesis of gastric MALT lymphoma (Table 1). The chemokine receptor, CXCR3, is expressed on activated T cells and B lymphoma cells [50]. Two studies revealed that CXCR3 and its ligand Mig were detected in the peripheral blood and lymphoma cells of the patients with gastric MALT lymphoma, suggesting that CXCR3 may be involved in the pathogenesis of neoplastic B cells that easily disseminate to the circulating systems and peripheral lymphoid organs [51,52]. Yamamoto et al. further found that the CXCR3 expression in tumor cells of gastric MALT lymphoma is closely associated with the HP-negative infection, advanced-stage disease, and Baculovirus IAP repeat-containing 2 (API2)-MALT1 fusion proteins of the chromosome translation of t(11;18)(q21;q21), indicating that CXCR3 is not associated with the HP-dependent lymphomagenesis of gastric MALT lymphoma [53]. Deutsch reported that C-C motif chemokine receptors 7 (CCR7), CXCR3, and CXCR7 were more frequently detected in lymphoma cells of HP-positive gastric MALT lymphoma than in inflammatory cells of HP-positive gastritis [54]. When compared with gastric MALT lymphoma, the expression of CCR1, CCR5, CCR7, CCR8, CCR9, CXCR3, CXCR6, CXCR7, and X-C motif chemokine receptors 1 (XCR1) was higher in gastric diffuse large B cell lymphoma (DLBCL) with a histologic evidence of MALT (DLBCL[MALT]) [54]. Notably, CXCR4 was frequently detected in nodal MALT lymphoma, and nodal DLBCL patients with a bone marrow infiltration, but was not expressed in the tumor cells of patients with gastric lymphoma [54]. However, Stollberg et al. revealed that the CXCR4 expression was detected in 92% of the 55 cases of gastric and extragastric MALT lymphoma, and the expression of CXCR4 correlated with the proliferation index of Ki-67 [55]. In addition, the expressions of somatostatin receptors 3, 4, and 5 were more frequently observed in the gastric MALT lymphoma than in the extragastric MALT lymphoma [55]. In a murine model, Wang et al. found that the use of CXCR7 inhibitors resulted in the destruction of the architecture, and a reduction in the numbers of splenic marginal zone B cells [56]. Considering that CXCL12 is the ligand that interacts with CXCR4 and CXCR7, these findings suggested that during a HP infection, CXCL12 regulates the homing B cell lymphocytes to gastric mucosa, and further promotes the abnormal growth of B lymphocytes through CXCL12/CXCR4 or CXCL12/CXCR7 signaling.

HP colonization induces several innate and adaptive immune responses in the microenvironments of the stomach [57]. The CD4^+^CD25^+^ regulatory T cells (Tregs) have been shown to regulate the adaptive immune response for HP infection, and thus, to contribute to the persistent colonization of HP, and the pathogenesis of HP-related inflammation and its-related diseases [58,59]. Tumor-infiltrating Tregs in B-cell lymphoma are memory T cells that often express the forkhead box transcription factor FOXP3, a master gene involved in the regulation of the Treg cell lineage and a specific Treg marker [60]. In addition to suppressing the function and proliferation of effector T-cells in DLBCL, these intratumoral Tregs are capable of inhibiting the cytokine production (IFN-γ and IL-4) [61]. In a murine model of *H. felis*-induced gastric MALT lymphoma, Craig et al. showed that the development of MALT lymphoma requires both BCR signaling through the poly-reactivation of tumor Ig with certain antigens and tumor infiltrating T-cells [62]. Most of the tumor-infiltrating CD4+ cells in gastric MALT lymphoma were shown to be FOXP3+ Tregs, and these Tregs were recruited by tumor cells through the chemokines CCL17 and CCL22, secreted by FOXP3+ Tregs (Table 1) [62]. In addition, the systemic depletion of FOXP3+Tregs in vivo efficiently resulted in the regression of MALT lymphoma [62]. When compared with human gastritis, the expression of FOXP3+ Tregs, CCL17, and CCL22 was more frequently found in human MALT lymphoma samples [62]. Garcia et al. showed that HP-positive gastric MALT lymphomas contained higher FOXP3^+^/CD3^+^ cell ratios than HP-negative gastric MALT lymphomas [63]. Among the patients receiving first-line HPE, the median numbers of FOXP3+ and the FOXP3^+^/CD3^+^ cell ratios were higher in HP-dependent tumors than in HP-independent tumors [63]. In HP-dependent tumors, decreased FOXP3+ cells were found in samples that underwent HPE [63]. Iwaya et al. also showed that the number of FOXP3^+^ cells was higher in HP-positive gastric MALT lymphoma when compared with that of HP-negative MALT lymphoma and chronic gastritis [64]. Furthermore, HP-dependent tumors harbored more FOXP3^+^ cells and higher FOXP3^+^/CD4^+^ cell ratios than HP-independent tumors [64].

## 4. Epigenetic Changes and Genetic Changes Are Involved in the Lymphomagenesis of Gastric MALT Lymphoma

Epigenetic modifications both at the chromatin and DNA level affect the structure and the expression of genes [65]. The most widely studied epigenetic modification is the cytosine methylation in the context of the dinucleotide CpG [66,67]. Previous studies have demonstrated that HP infection causes DNA methylation and hypermethylation of the CpG islands, which alter the promoter regions of tumor suppressor genes and result in the deletion and loss of expression of these genes. The silenced tumor suppressor genes further drive these cells to undergo malignant transformations through the abnormal promotion of cell proliferation and cell cycle alterations [68,69,70]. For gastric lymphoma, several studies have reported that the methylation of p16^INK4A^, an inhibitor of cyclin-dependent kinases, may be associated with the development of gastric MALT lymphomas [71,72,73]. Kim et al. reported that p16 hypermethylation was detected in seven (58%) of 12 patients with HP-dependent gastric MALT lymphoma [71]. Min et al. revealed that the methylation of p57^KIP2^ was detected in HP-positive gastric MALT lymphomas but not in HP-positive gastric lymphoid follicles [72]. Park et al. found that the methylation of p16^INK4A^ was detected in 30 (75%) of 40 gastric MALT lymphoma patients, and that the p16^INK4A^ methylation correlated with the negative chromosome translation of t(11;18)(q21;q21) [73]. Two studies have reported the relationship between the CpG island methylator phenotype (CIMP) and developments of gastric MALT lymphoma [74,75]. Kaneko et al. found that CIMPs (*p16*, *hMLH1*, *MINT1*, *MINT2*, and *MINT31*) are more frequently detected in HP-dependent gastric MALT lymphoma than in HP-independent tumors [74]. Kondo et al. also found that the frequency of CIMP was higher in HP-positive gastric lymphomas than in HP-negative gastric lymphomas, and that the aberrant CpG hypermethylation of specific target genes, including *p16*, *MGMT*, and *MINT31*, correlated with the HP infection-associated lymphoma [75]. These findings indicated that HP might cause the aberrant methylation of DNA-specific genes (*p16* and *p57*) and CpG island-specific genes (*hMLH1*, *MINT1*, *MINT2*, and *MINT31)*, which are important epigenetic mechanisms contributing to the HP-dependent lymphomagenesis of gastric MALT lymphoma (Table 1). In contrast to the epigenetic change-related genes of gastric MALT lymphoma, affected genes resulting from the HP-induced DNA methylation in the carcinogenesis of gastric cancer comprise cell adhesion pathway-regulated genes (*CDH1*, *VEZT*, *CX32*, and *CX43*), cell cycle regulation-related genes (*CDKN2A,* encoding the p16^INK4A^ protein), DNA mismatch repair genes (*MHL1* and *MGMT*), inflammation-related genes (*TFF2* and *COX-2*), transcriptional factors-encoding genes (*RUNX3*, *FOXD3*, *USF1*, *USF2*, *GATA4*, and *GATA5*), autophagy-related genes (*ATG16L1* and *MAP1LC3A*), and most tumor suppressor genes (*LOX*, *HRASLS*, *THBD*, *HAND1*, *FLN*, *p41ARC*, *WWOX*, *CYLD*, and *PTEN*) [76].

Antigenic stimulation by HP is regarded as a major inducer of HP-related carcinogenesis in the stomach [77]. However, only <0.01% of the HP-infected patients eventually develop gastric MALT lymphoma [77]. We have previously reported that polymorphisms in *TNF-α*, *GSTT1*, and *CTLA4* genes are associated with the risk of gastric MALT lymphoma [78,79,80], indicating that the host factors may be susceptible to the development of this rare lymphoma. Furthermore, we assessed the genetic variation of the cytokines and chemokines associated with gastrointestinal inflammation and with immunity in gastric MALT lymphoma cases and unrelated controls, and found that five single-nucleotide polymorphisms of IL-22 were associated with the susceptibility to gastric MALT lymphoma [81]. We also found that co-culturing HP with peripheral mononuclear cells or CD4(+) T cells in gastric epithelial cells stimulated the secretion of IL-22, which upregulated the expression of the two antimicrobial proteins RegIIIα and lipocalin-2 [81]. Furthermore, in 41 gastric MALT lymphoma patients undergoing HPE, the expression of IL-22 was significantly associated with the HP-dependence of gastric MALT lymphoma [81]. Moreover, we demonstrated that in gastric epithelia-derived AGS (human gastric epithelial adenocarcinoma cell line; American Type Culture Collection CRL 1739) cells, HP significantly induced the CCL20 expression through the activation of nuclear factor (NF)-κB, whereas IL-22 inhibited the induction of CCL20 by attenuating NF-κB activation [82]. The knockdown of endogenous signal transducer and activator of transcription 3 (STAT3) significantly reduced the inhibitory effect of IL-22 [82]. In gastric MALT lymphoma samples, the CCL20 expression significantly correlated with the loss of HP dependence [82]. These findings indicated that the susceptibility of gastric MALT lymphoma is influenced by genetic polymorphisms in IL-22, and that the product of IL-22 is involved in the mucosal immunity against HP and is associated with the tumor response to HPE (Table 1).

## 5. The Direct Evidence of Cytotoxin-Associated Gene A in the Lymphomagenesis of Gastric MALT Lymphoma

The cytotoxin-associated gene A (CagA) protein, the most important HP virulence factor, consists of an amino-terminal domain, central domain, N-terminal binding sequence-located domain, and a C-terminal unstructured region [83,84,85]. The C-terminal domain contains a 5-amino acid repetitive tandem motif, glutamic acid-proline-isoleucine-tyrosine-alanine (EPIYA), which is the crucial site for the tyrosine phosphorylation of CagA [83,84,85]. Epidemiologic studies have shown that the presence of anti-CagA antibodies is linked to the risk of formation of lymphoid follicles and lymphoma in the stomach [86,87]. Eck et al. revealed that serum immunoglobulin G antibodies to CagA were found in 95.5% of HP-seropositive gastric MALT lymphoma cases and in 67% of HP-seropositive chronic active gastritis cases [88]. Among t(11;18)(q21;q21)-negative gastric MALT lymphoma cases, the titers of anti-CagA were significantly higher in HP-dependent cases than in HP-independent cases [89]. These findings suggested that HP strains expressing the CagA protein may be associated with the lymphomagenesis of gastric MALT lymphoma.

In gastric cancer models, CagA passes from the HP cytosol to gastric epithelial cells using the type IV secretion system (T4SS) and further undergoes tyrosine phosphorylation by Src and abelson murine leukemia viral oncogene homolog (Abl)-family kinases on specific tyrosine residues within the EPIYA motif [83,84,85]. Then, the phosphorylated CagA binds to SHP-2 and causes an aberrant activation of signaling of the extracellular signal-regulated kinase (ERK) and the p38 mitogen-activated protein kinase (MAPK) [83,84,85]. In addition to the promotion of the tumorigenesis of gastric cancer, CagA is able to translocate into human B-lymphocytes and further upregulates ERK1/2 phosphorylation and its downstream phosphorylation of Bad at Ser^112^, which further inhibits the apoptosis of B-lymphocytes [90]. In CagA-transgenic mice, Ohnishi found that CagA can promote the development of gastrointestinal and hematological malignancies through CagA-regulated the tyrosine phosphorylation and the subsequent SHP-2-deregulation [28]. Our group demonstrated that CagA can be directly translocated to human B lymphoid cells in vitro via the T4SS encoded by the *cag* pathogenicity island (*cag*PAI), and intracellular CagA co-immunoprecipitates with SHP-2 after tyrosine phosphorylation [27]. The interactions between CagA and phospho (p)-SHP-2 further promoted the proliferation and inhibited the apoptosis of B-lymphocytes through the activation of ERK and p38 MAPK and the upregulation of Bcl-2 and Bcl-xL [27]. The clinical and biological significance of CagA in B-lymphocytes was further demonstrated in tumor samples of our 64 stage IE gastric MALT lymphoma patients, in which the CagA expression was higher in HP-dependent tumors (26/38 (68.4%)) than in HP-independent tumors (5/26 (19.2%), *p* < 0.001) [17]. Our results also showed that the CagA expression was closely associated with the quick response to HPE in HP-dependent cases (median time for CR after completion of HPE; 3.0 months versus 6.5 months, respectively) [17]. Furthermore, we revealed that CagA translocation is highly associated with the expression of p-SHP-2, p-ERK, p-p38 MAPK, Bcl-2, and Bcl-xL in gastric MALT lymphoma, and that the expressions of these molecules were associated with HP-dependence [91]. When compared with the CagA expression alone, our results showed that the combined expressions of CagA, p-SHP-2, and p-ERK raised the positive predictive value as well as the specificity for HP-dependence of gastric MALT lymphoma [91] (Table 1). 

In contrast to the CagA activity in the deregulation of intracellular signaling pathways in B-lymphocytes through tyrosine phosphorylation-dependent SHP-2 signaling, CagA can stimulate AKT Serine/Threonine Kinase 1 (AKT1) and upregulate human homolog of double minute 2 (HDM2), and subsequently impair p53 function that can inhibit the apoptosis of B-lymphocytes [92]. In interleukin 3-dependent B cells, CagA suppressed the cell proliferation via the inhibition of the Janus kinase (JAK)-STAT signaling pathway during the G1-S transition [92]. However, the role of JAK-STAT-signaling in CagA-induced gastric MALT lymphoma has been unclear. Taken together, the aforementioned evidences indicated the involvement of CagA in the lymphomagenesis of gastric MALT lymphoma: (1) CagA deregulates the intracellular signaling pathways in B-lymphocytes through a tyrosine phosphorylation-dependent signaling pathway to initiate the molecular pathogenesis of this tumor. The signaling pathway from the CagA tyrosine phosphorylation (SHP-2, ERK, BAD, p38 MAPK (mitogen-activated protein kinase), Bcl-2, and Bcl-xL) prevents human B lymphocytes from apoptosis, allowing the lymphocytes to acquire a survival ability and develop hematological malignancies. (2) Conversely, the signaling pathway from the CagA tyrosine phosphorylation-independent pathway, such as the impairment of p53 that can inhibit the apoptosis of B-lymphocytes, and the inhibition of JAK/STAT by IL-3 signaling, results in the suppression of proliferation of B-lymphocytes. These findings indicate that the inequities between apoptosis and proliferation resulted from differential CagA-related signaling that is involved in the lymphomagenesis of HP-dependent gastric MALT lymphoma (Figure 1).

## 6. CagA and Its Regulated Immune Response in Gastric Microenvironment May Participate in the Lymphomagenesis of Gastric MALT Lymphoma

Our in vitro study showed that during the co-culturing of HP with B-lymphocytes cells, a higher upregulation of CD86 was stimulated by a CagA wild-type HP strain than by a CagA-deficient HP strain [27]. Previous studies have suggested that CagA plays an important role in the migration of HP-primed CD4+ T cells and in the induction of differentiation of Tregs [93]. In a study on the relationship between the expression of CD4 and FOXP3 in patients with peptic ulcer, Bagheri et al. showed that the number of CD4^+^ T cells positively correlated with CagA, whereas FOXP3^+^ T cells as well as the expressions of IL-10, TGF-β1, FOXP3, and INF-γ were closely associated with the density of HP [94]. Lina et al. reported that HP T4SS components activate the p38 MAPK pathway, upregulate the B7-H1 (PD-L1) expression, and cause the FOXP3^+^Treg cell induction in gastric epithelial cells [95]. Another study showed that phosphorylation of the HP CagA in macrophages can enhance M1/Th1/Th17 responses, decrease the regulatory macrophage response, and reduce HP colonization [96].

The CD56 (+) natural killer (NK) cell, an important component of the innate lymphoid cells (ILCs), has the ability to modulate both humoral- and cell-mediated immune responses [97,98]. The HP infection enhances the production of Th2 cytokines (IL-10) that can promote the proliferation of CD56 in a gastric microenvironment [99]. The infiltration of NK cells is also detected in gastric MALT lymphoma [99]. Since CD56+ NK cells restrict the extent of HP-related autoreactive and neoplastic B lymphoid cells in the stomach, this finding may explain that the increased infiltrations of NK cells closely correlated with the HP-dependence of early-stage gastric DLBCL(MALT) [100]. 

In addition to NK cells, ILCs, lacking immune cell lineage, are divided into: (1) group 1 ILCs (ILC1s), produce IFN-γ, and require Tβ help; (2) group 2 ILCs (ILC2s), produce type 2 cytokines like IL-5 and IL-13, and require GATA3 help; and (3) group 3 ILCs (ILC3s), produce IL-17 and/or IL-22, and require retinoic acid related orphan receptor (ROR)γt help, based on the distinct production of cytokines and the transcriptional factors-dependent help [98,101,102]. Previous studies suggested that persistent HP infection can allow the recruitment of the ILCs to gastric mucosa, which usually lack lymphoid tissue, and can promote the development of gut-associated lymphoid tissue [103,104]. Several studies with the *H*. species-infected animal models have demonstrated that IFN-γ can induce the formation of MALT, a precursor lesion of gastric MALT lymphoma [105,106]. However, IFN-γ also exhibits antitumor effects by producing NK cells, differentiating CD4 cells to Th1 cells, and activating the cytotoxicity of CD8+ cells [107,108]. Mucosa-associated invariant T (MAIT) cells, a type of antimicrobial innate T cells, are expressed in a variety of organs, including the blood, liver, lung, intestine, and stomach [109]. In the gastric mucosa of both patients and mice, D’Souza et al. showed that HP infection enhanced the production of MAIT cells [110]. Booth et al. found that MAIT cells were frequently lower in the blood of HP-positive gastritis patients than in the blood of HP-negative patients, whereas the expression of MAIT cells in gastric tissue was not different between HP-positive and HP-negative patients [111]. Using an in vitro cell study, they revealed that MAIT triggered by HP-infected macrophages harbored abilities of cytotoxicity through the production of IFN-γ, TNF-α, and IL-17 [111]. These findings indicate that ILCs are not only involved in the early process of lymphomagenesis of gastric MALT lymphoma but also limit the autonomous growth and progression of the lymphoma cells through the production of IFN-γ, NK cells, and IL-22 and through the interaction with adaptive immune cells. These hypotheses are supported by our findings that the IL-22 expression significantly correlated with the HP-dependent status of gastric MALT lymphoma [81]. 

A proliferation inducing ligand (APRIL), a member of the tumor necrosis factor (TNF) family, is involved in the pathogenesis of B cell neoplasms [112,113]. Munari et al. showed that APRILs in gastric MALT lymphoma samples were released by tumor-infiltrating macrophages [114]. Recently, Floch et al. demonstrated that the infiltrations of leukocytes (most B cells and few CD4^+^ T cells) were predominant in APRIL transgenic mice than in wild-type mice infected by *Helicobacter* species [115]. Importantly, these B cells expressed marginal zone origin-like B surface markers [115].

Yokoyama et al. reported that CagA inhibits the progression of the cell cycle through the activation of a nuclear factor of activated T cell (NFAT) and NFAT-dependent genes, such as *p21* in gastric epithelial cells [116]. The activating NFAT, triggered by calcium-dependent serine/threonine phosphatase calcineurin signaling, has been demonstrated to be involved in the lymphomagenesis of a variety of lymphomas [117,118]. NFATc1 (also known as NFAT2), a member of the NFAT family, has been detected in tumor samples of B-cell lymphoid neoplasms, including MALT lymphoma, Burkitt’s lymphoma, and Hodgkin’s lymphoma [119,120]. In a type I Burkitt’s lymphoma cell line study, Kondo et al. showed that NFAT/calcineurin signaling promoted the BCR-signaling-mediated apoptosis of B-lymphocytes [121]. Our preliminary results showed that in HP-co-cultured lymphoma cells, CagA upregulated the expressions of p-SHP-2, p-ERK, and Bcl-xL, and nuclear NFATc1 translocation [122]. The HP-co-cultured lymphoma cells exhibited a G1 phase suppression through the activation of NFATc1 and p21. We further found that the nuclear NFATc1 expression was significantly higher in HP-dependent than in HP-independent tumors, and closely correlated with the expression of CagA (Table 1) [122]. The NFAT signaling pathway also regulates the immune responses of FOXP3+ Tregs as well as Th17 helper cells-related cytokines (IL-22) [57].

Taken together, these findings suggested that the lack of tumor regression after HPE may be associated with a change in the immunological microenvironment, such as the decrease in the activity of CD56 (+) NK cells, lack of macrophage activity, and decrease in the expression of NFATc1 (Figure 2).

## 7. Microbiomes May Be Associated with Antibiotic-Responsiveness of Gastric MALT Lymphoma

Although HP-specific intratumor T-cells, Tregs, and their communication with B-cells, or the link antigen stimulation via BCR signaling or HP CagA strain-regulated signaling, are associated with the HP-dependence of gastric MALT lymphoma [4,25,41,123,124,125], the aforementioned effects are not detected in certain proportions of HP-dependent gastric MALT lymphoma patients. In addition to HP-positive gastric MALT lymphoma, we and other investigators have demonstrated that around 30% of HP-negative gastric MALT lymphoma are still responsive to a first-line antibiotic treatment [21,22,23,24]. We also revealed the absence of the *CagA* gene in gastric tumor biopsies obtained from HP-negative patients with antibiotic-responsive tumors [24]. These findings suggested the existence of other molecular mechanisms in the lymphomagenesis of these antibiotic-responsive HP-negative gastric MALT lymphoma patients.

Previous studies have reported that gastric MALT lymphoma may arise from patients infected with *H. heilmannii* that is not identified from the routine test for HP infection [126,127,128]. Like HP-positive gastric MALT lymphoma patients, HPE-like regimens also eradicate *H. heilmannii* and cause the regression of *H. heilmannii*-associated gastric MALT lymphoma [126,127,128]. In addition to the association between *H. heilmannii* and the lymphomagenesis of MALT lymphoma in human and animal models [126,127,128,129], BALB/c mice infected with *H. felis* developed MALT-type lymphoma and these lymphomas could be regressed after an antibiotic treatment [130,131]. These findings indicated that *Helicobacter*-like bacteria are also involved in the lymphomagenesis of some antibiotic-responsive HP-negative gastric MALT lymphoma patients.

However, we cannot exclude the possibility that the HPE regimen may also eradicate other bacteria or gut microbiota that may be involved in the lymphomagenesis of antibiotic-responsive HP-negative gastric MALT lymphomas. In fact, certain bacteria, like *Campylobacter jejuni*, *Borrelia bergdorferi*, *Chlamidia psitacci,* and *Achromobacter xylosoxidans* are associated with the development of extragastric MALT lymphomas [132,133]. Two studies reported that *Haemophilus influenza* and *Staphylococcal enterotoxin A* may be etiologically linked with pediatric nodal marginal zone B-cell lymphoma and cutaneous T-cell lymphoma, respectively [134,135].

In addition to the direct role of microbiota in the initiation of the lymphomagenesis of gastric MALT lymphoma, gut microbiota may alter the immune response and immune parameters [136,137,138,139,140]. For example, mice colonized with distinct communities of resident bacteria can deplete marginal zone B cells using CD8+ cytotoxic T cells [139]. In another study using an ataxia telangiectasia transgenic (*atm*^−/−^) mice model, Yamamoto et al. found that *atm*^−/−^ mice with restricted gut microbiota exhibited a delayed incidence of lymphoma and had a longer lifespan than *atm*^−/−^ mice with conventional microbiota [140]. In their study, the inoculation of *Lactobacillus johnsonii* isolated from restricted gut microbiota reduced the systemic inflammation (decreased the levels of IL-1b and IFN-γ), molecular oxidative stress, and systemic leukocyte genotoxicity [140]. Previous studies have also demonstrated that *Lactobacillus johnsonii* can alleviate HP-associated inflammation (the lymphocytic and neutrophilic infiltration in the lamina propria) as well as gastritis [141,142,143]. These findings suggested that intestinal microbiota may be involved in the lymphomagenesis of B-cell lymphoma by deregulating the immune response, inflammation, and genotoxicity.

## 8. Conclusions

Although our findings and those of other studies suggested that CagA-related signal transduction can directly affect the lymphomagenesis of HP-dependent gastric MALT lymphomas, other HP-related pathways, such as tumor-infiltrating T lymphocytes, cytokines, chemokines, FOXP3+Tregs, the communication of co-stimulatory molecules, and the HP-antigen-triggering BCR pathway, are also involved in the lymphomagenesis of HP-dependent gastric MALT lymphomas. We showed that CagA upregulated CagA-related signaling molecules and the NFAT transcription factor in B-lymphocytes. In addition to HP-associated lymphoma, HP-like bacteria or other bacteria are associated with the lymphomagenesis of antibiotic-responsive HP-negative gastric MALT lymphoma. Based on these findings, we propose that the development of HP-dependent MALT lymphoma cells in the stomach requires at least four signals for cognate help, as summarized in Figure 2. Considering the potential immune reaction resulting from the microbiota, further explorations on the precise mechanisms of HP-unrelated microbiota in the lymphomagenesis of antibiotic-responsive HP-negative gastric MALT lymphoma are merited.

## Figures and Tables

**Figure 1 cancers-11-00547-f001:**
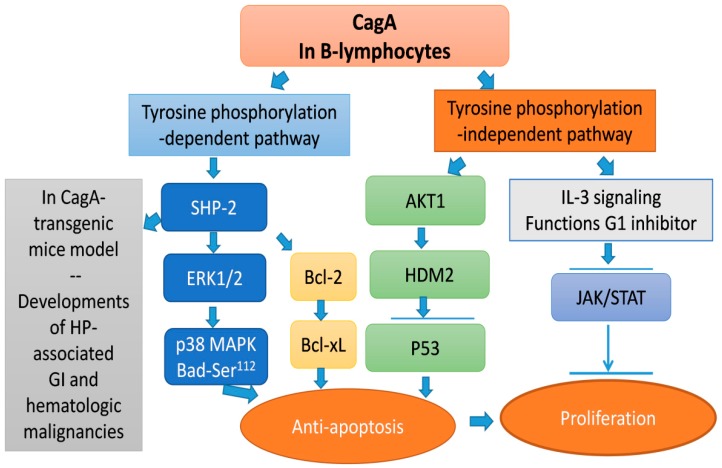
The cytotoxin-associated gene A (CagA) deregulates intracellular signaling pathways in B-lymphocytes in tyrosine phosphorylation-dependent and -independent manners to initiate the lymphomagenesis of *Helicobacter pylori* (HP)-dependent gastric mucosa-associated lymphoid tissue (MALT) lymphoma. The HP protein CagA was translocated into B lymphocytes and co-immunoprecipitated with phospho-Src homology-2 domain-containing phosphatase (SHP-2), and further activated extracellular signal-regulated kinase (ERK), p38 mitogen-activated protein kinase (MAPK), B cell lymphoma (Bcl)-2-associated death promoter (BAD), Bcl-2, and Bcl-xL. The tyrosine phosphorylation-dependent signaling pathway promotes the proliferation and inhibits the apoptosis of B lymphocytes, allowing these B-lymphocytes to acquire an oncogenic survival ability and develop hematological malignancies. In contrast, the signaling pathways from the CagA tyrosine phosphorylation-independent pathway, including the impairment of p53 and inhibition of Janus kinase (JAK)/STAT, resulted in the suppression of apoptosis and the inhibition of proliferation of B-lymphocytes, respectively. The imbalance between apoptosis and proliferation leads to the pathogenesis and development of HP-dependent gastric MALT lymphoma.

**Figure 2 cancers-11-00547-f002:**
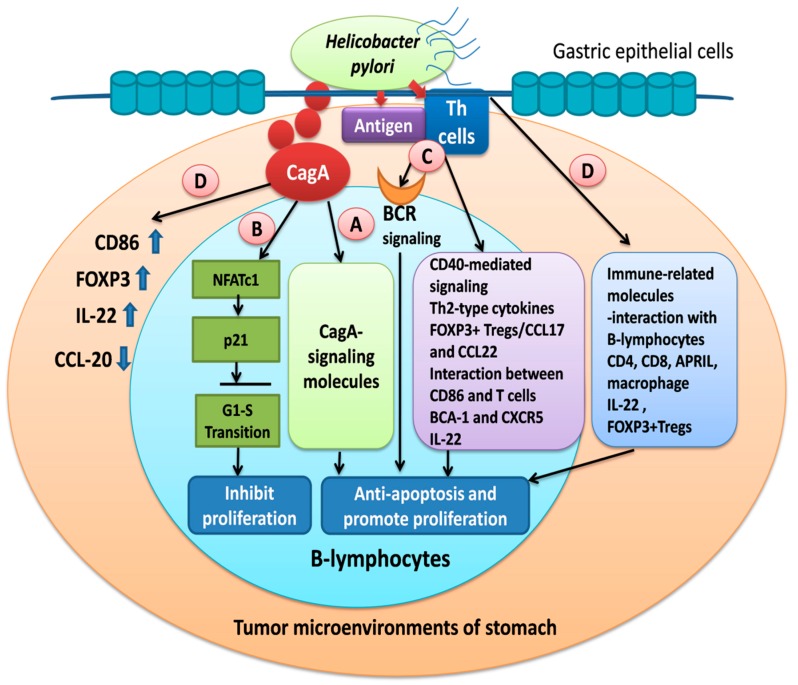
Involvement of ctotoxin-associated gene (CagA)-derived signals, T-cell-derived signals, and tumor microenvironment-related mediators in the *Helicobacter pylori* (HP)-induced lymphomagenesis of gastric mucosa-associated lymphoid tissue (MALT) lymphoma. HP infection stimulates T lymphocytes and cytokines in the gastric mucosa and indirectly induces the development of MALT, from which B lymphocytes migrate and infiltrate the site of MALT in the stomach. (**A**) The CagA protein translocates into B lymphocytes as it is secreted by HP on gastric epithelial surfaces through the T4SS system, thereby resulting in a cascade of survival signaling in the B lymphocytes. (**B**) Simultaneously, CagA restricts the proliferation of these lymphoma B cells by upregulating the G1 cell-cycle phase regulator, p21, by triggering nuclear factor of activated T cell (NFAT) c1 signaling. (**C**) HP infection also indirectly promotes lymphomagenesis through T-cell-stimulatory pathways, such as CD40-mediated signaling, Th-2-type cytokine-mediated signaling, and forkhead box P3 (FOXP3)+ regulatory T cells (Tregs) and B-cell receptor (BCR)-signaling that mediate the interaction between co-stimulatory molecules such as CD86 and CTLA4 in T-lymphocytes. (**D**) Molecular cross-talk between lymphoma B cells and immune-associated molecules in the tumor microenvironment (T cells, FOXP3+ Tregs, and Th17 helper cell-regulated cytokines including interleukin (IL)-22, chemokines, and chemokine receptors, and the interaction between macrophage and a proliferation-inducing ligand (APRIL)) stimulates the survival signaling in B-lymphocytes.

**Table 1 cancers-11-00547-t001:** Biologic markers are associated with the antibiotics-responsiveness of gastric MALT lymphoma patients who received a first-line antibiotics treatment.

Indirect and Direct Lymphomagenesis-Related Signaling	Makers	Methods	Reference
Intratumor T cells	CD40/CD40L	IHC	25,31
Co-stimulatory molecules	CD86 (B7.2)	IHC	40,100
CD4^+^CD56^+^ regulatory T-cell	FOXP3	IHC	62,63,64
Chemokine receptor	CCL17 and CCL20	IHC	62
Chemokines and receptor	BCA-1/CXCR5	IHC	49
Chemokines	IL-22	IHC	81
Methylation	p16^INK4A^	Methylation-Specific PCR	71,73
Natural killer cell	CD56	IHC	100
HP-specific protein	CagA protein	IHC	17,27
HP-specific protein	Serum CagA IgG antibody	ELISA (a CagA kit)	89
CagA-signaling molecules	p-SHP-2, p-ERK, Bcl-2	IHC	91
CagA-signaling molecules	NFATc1	IHC	119,120,122

HP: *Helicobacter pylori*; IHC: immunohistochemistry; PCR: polymerase chain reaction; ELISA: enzyme-linked immunosorbent assay; FOXP3: forkhead box P3; CCL: chemokine ligands; CXCR: CXC chemokine receptor; IL: interleukin; CagA: cytotoxin-associated gene A; IgG: Immunoglobulin G; SHP-2: phospho-Src homology-2 domain-containing phosphatase; ERK: extracellular signal-regulated kinase; Bcl: B cell lymphoma; NFAT: nuclear factor of activated T cell.

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
