# Peer review of "Novel Insights of Lymphomagenesis of Helicobacter pylori-Dependent Gastric Mucosa-Associated Lymphoid Tissue Lymphoma"

_cancers, 2019, doi:10.3390/cancers11040547_

Round 1

Reviewer 1 Report

)In this review, authors have summarized the mechanisms of involvement of T-cell-derived signals and CagA-triggering signals in the HP-dependent lymphomagenesis of HP-positive gastric MALT lymphoma. Authors showed that CagA upregulated the CagA-related signaling molecules and NFAT transcription factor in B-lymphocytes. Authors propose that the development of HP-dependent MALT lymphoma cells of stomach requires at least four signals for proliferation. Overall the study well organized and described in a logical way. I suggest the following minor additions

Add more details about epigenetic mechanism affecting gene expression and leading to lymphomagenesis, and explain if it is similar of different from HP induced epigenetic changes leading to gastric carcinogenesis (Pathogens 20198(1), 23). 

A brief section regarding methodology would be great, mentioning the databases searched and how the studies were selected to be included in this study and what basis they were excluded.

Author Response

Comments from Reviewer 1

In this review, authors have summarized the mechanisms of involvement of T-cell-derived signals and CagA-triggering signals in the HP-dependent lymphomagenesis of HP-positive gastric MALT lymphoma. Authors showed that CagA upregulated the CagA-related signaling molecules and NFAT transcription factor in B-lymphocytes. Authors propose that the development of HP-dependent MALT lymphoma cells of stomach requires at least four signals for proliferation. Overall the study well organized and described in a logical way. I suggest the following minor additions

1.       Add more details about epigenetic mechanism affecting gene expression and leading to lymphomagenesis, and explain if it is similar of different from HP induced epigenetic changes leading to gastric carcinogenesis (Pathogens 20198(1), 23). 

Authors’ response:

We thank you for this pertinent suggestion. Accordingly, we have provided more information on the epigenetic mechanisms associated with gastric MALT lymphoma (Page 6, Lines 247-252; Lines 255-257; Lines 262-276).

Previous studies have demonstrated that HP infection causes DNA methylation and hypermethylation of the CpG islands, which alter the promoter regions of tumor suppressor genes and result in the deletion and loss of expression of these genes. The silenced tumor suppressor genes further drive these cells to undergo malignant transformations through the abnormal promotion of cell proliferation and cell cycle alterations [68-70]. Min et al. revealed that methylation of p57KIP2 was detected in HP-positive gastric MALT lymphomas but not in HP-positive gastric lymphoid follicles [72].  Kondo et al. also found that the frequency of CIMP was higher in HP-positive gastric lymphomas than in HP-negative gastric lymphomas, and aberrant CpG hypermethylation of specific target genes, including p16, MGMT, and MINT31, correlated with the HP infection-associated lymphoma [75]. These findings indicated that HP might cause aberrant methylation of DNA-specific genes (p16 and p57) and CpG island-specific genes (hMLH1, MINT1, MINT2, and MINT31), which are important epigenetic mechanisms contributing to HP-dependent lymphomagenesis of gastric MALT lymphoma (Table 1). In contrast to the epigenetic change-related genes of gastric MALT lymphoma, affected genes resulting from HP-induced DNA methylation in the carcinogenesis of gastric cancer comprise cell adhesion pathway-regulated genes (CDH1, VEZT, CX32, and CX43),  cell cycle regulation-related genes (CDKN2A,  encoding p16INK4A protein),  DNA mismatch repair genes (MHL1 and MGMT),  inflammation-related genes (TFF2 and COX-2), transcriptional factors-encoding genes (RUNX3, FOXD3, USF1, USF2, GATA4, and GATA5), autophagy-related genes (ATG16L1 and MAP1LC3A), and most tumor suppressor genes (LOX, HRASLS, THBD, HAND1, FLN, p41ARC, WWOX, CYLD, and PTEN) [76].

References

68.     Jones, P.A.; Takai, D. The role of DNA methylation in mammalian epigenetics. Science 2001, 293, 1068–1070.

69.     Perri, F.; Cotugno, R.; Piepoli, A.; Merla, A.; Quitadamo, M.; Gentile, A.; Pilotto, A.; Annese, V.; Andriulli, A. Aberrant DNA methylation in non-neoplastic gastric mucosa of H. pylori infected patients and effect of eradication. Am. J. Gastroenterol. 2007, 102, 1361.

70.     Ando, T.; Yoshida, T.; Enomoto, S.; Asada, K.; Tatematsu, M.; Ichinose, M.; Sugiyama, T.; Ushijima, T. DNA methylation of microRNA genes in gastric mucosae of gastric cancer patients: its possible involvement in the formation of epigenetic field defect. Int. J. Cancer 2009, 124, 2367–2374.

72.     Min, K.O.; Seo, E.J.; Kwon, H.J.; Lee, E.J.; Kim, W.I.; Kang, C.S.; Kim, K.M. Methylation of p16(INK4A) and p57(KIP2) are involved in the development and progression of gastric MALT lymphomas. Mod. Pathol. 2006, 19, 141–148.

75.  Kondo, T.; Oka, T.; Sato, H.; Shinnou, Y.; Washio, K.; Takano, M.; Morito, T.; Takata, K.; Ohara, N.; Ouchida, M.; et al. Accumulation of aberrant CpG hypermethylation by Helicobacter pylori infection promotes development and progression of gastric MALT lymphoma. Int. J. Oncol. 2009, 35, 547–557.

76.     Muhammad, J.S.; Eladl, M.A.; Khoder, G. Helicobacter pylori-induced DNA Methylation as an Epigenetic Modulator of Gastric Cancer: Recent Outcomes and Future Direction. Pathogens. 2019, 8, pii: E23.

77. A brief section regarding methodology would be great, mentioning the databases searched and how the studies were selected to be included in this study and what basis they were excluded.

Authors’ response:

We greatly appreciate your comment, and as per your suggestion, we have provided detailed information about the inclusion and exclusion criteria for the studies of gastric MALT lymphoma (Page 3, Lines 121-139).

The approaches of literature survey described in this review article use a combination of publicly available databases PubMed, Ovid, SCOPUS, and Web of Science. In the literature survey, we have used the following key terms “B cell lineage”,  “histological manifestation”, “Immunophenotype”, “T cells”, “cytokines”, “chemokine”, “chemokine receptors”, “regulatory T cells (FOXP3)”, “epigenetic”, “genetic”, “DNA methylation”, “cytotoxin-associated genes”, “natural killer cell”, “macrophage”, “innate lymphoid cells”, “mucosa-associated invariant T”, “microbiomes”, and “microbiota”, in combination with “Helicobacter pylori”, “mucosa-associated lymphoid tissue lymphoma”, “gastric lymphoma”, and “gastrointestinal lymphoma”. In addition, we have used “Helicobacter helimannii”, “Helicobacter felis”, in combination with  “mucosa-associated lymphoid tissue lymphoma”, and “gastric lymphoma”. However, we excluded “microRNA”, “antibiotics-unresponsive mechanisms-associated molecules, such as API2-MALT1, BCL10, and NF-κB”, genetic changes and chromosomes that contribute to HP-independence, advanced stages, and progression of gastric MALT lymphoma; for example, t(11;18)(q21;q21), t(1;14)(p22;q32), and t(14;18)(q32;q21) in this review. The results of the selected articles and our previously published reports were included in this study for reviewing the precise mechanisms of HP- and other bacteria-related, gastric microenvironment-related, epigenetic-related, or genetic-related, CagA-direct signaling-related pathogenesis of antibiotics-responsive gastric MALT lymphoma. The different mechanisms contributing to lymphomagenesis of antibiotics-responsive gastric MALT lymphoma are described under their specific subheadings.

Reviewer 2 Report

This is an interesting, well constructed and well written review which sets out the role of host and pathogen (H. pylori and some other Helicobacter species) associated factors in the etiology of tumors of the Mucosal Associated Lymphoid Tissue (MALT).

Overall the review is of interest in the readership of ‘Cancers’ and provides a contemporary overview of the field.

There are a few minor areas which would enhance the manuscript.

1.       Can the authors provide some detail/ comment on the memory B cell differentiation phenotype of the transformed B cells – (in addition to pan B cell antigen CD19, CD20, CD79a etc) and chemokine/ cytokine receptors identified in different studies and whether phenotypic differences relate to the extent of growth and invasion of the tumour. An overall comment should be made about the maturity of the B cell lineage involved.

2.       Can the authors cite references and/ or comment on, in addition to NK cells the potential involvement of tissue associated innate lymphoid cells which could, for example, act as sources of IL-22 and IFN-g (eg. MAIT cells, ILC1-3) in either promoting or restricting tumor growth.

Author Response

Comments from Reviewer 2

This is an interesting, well constructed and well written review which sets out the role of host and pathogen (H. pylori and some other Helicobacter species) associated factors in the etiology of tumors of the Mucosal Associated Lymphoid Tissue (MALT).

Overall the review is of interest in the readership of ‘Cancers’ and provides a contemporary overview of the field.

There are a few minor areas which would enhance the manuscript.

1.   Can the authors provide some detail/ comment on the memory B cell differentiation phenotype of the transformed B cells – (in addition to pan B cell antigen CD19, CD20, CD79a etc) and chemokine/ cytokine receptors identified in different studies and whether phenotypic differences relate to the extent of growth and invasion of the tumour. An overall comment should be made about the maturity of the B cell lineage involved.

Authors’ response:

We thank you for your insightful comments. As per your suggestion, we have provided more information on memory B cell and maturity of the B cell lineage involved in the lymphomagenesis of gastric MALT lymphoma (Page 2, Lines 58-78).

Tumor cells of gastric MALT lymphoma are located in the marginal zone and are surrounded by the reactive germinal center (GC), and these lymphoma cells are thought to arise from post-GC memory/marginal zone B cell [6,7]. During the B cell maturation process, the primary B cell repertoire generated in the bone marrow can differentiate to immature B cells (expression of a complete immunoglobulin (Ig)M-molecule) through multiple steps of rearrangement of Ig heavy chain and light chain genes [6-9]. Further, the immature B cells can transform to mature B cells (express both IgG and IgM) through alternative splicing of Ig heavy-chain mRNA [6-9]. These naïve mature B cells can migrate to the peripheral lymphoid organs or extranodal Peyer’s patches-like lymphoid follicles (MALT) and develop into short-lived plasma cells, or enter into the GC to transform centroblasts, following antigen stimulation [6,7]. In the GC, centroblasts can develop into centrocytes through somatic hypermutation and heavy chain class-switching recombination. Furthermore, selected GC B cells may differentiate into post-GC B cells, including plasma cells or memory/marginal zone B cells [6,7]. These neoplastic marginal zone memory B cells may resemble the centrocytes of GC, and share the common immunophenotype markers of normal marginal zone B cell, including pan-B antigen (CD19, CD20, CD22, and CD79), but do not express GC B cell profiles, including CD10 and B cell lymphoma (Bcl)-6 [1,2,5,7]. Neoplastic cells of gastric MALT lymphoma cells are positive for surface Igs (most IgM, and less frequently IgG and IgA), CD43, Bcl-2, and complement receptors (CD21, CD35) [1,2,5,7,10], whereas CD5, CD23, and cyclin D1 immunostains are generally negative in MALT lymphoma cells [1,2,5,7,10], although rare cases of CD5-positive MALT lymphomas have been reported [11].

In addition, we have provided more information on chemokine/cytokine receptors identified in different studies of gastric MALT lymphoma, and whether chemokine/cytokine receptors are associated with the extent of growth and invasion of the tumor (Page 5, Lines 198-214).

Deutsch reported that CCR7, CXCR3, and CXCR7 were more frequently detected in lymphoma cells of HP-positive gastric MALT lymphoma than in inflammatory cells of HP-positive gastritis [54]. When compared with gastric MALT lymphoma, the expression of CCR1, CCR5, CCR7, CCR8, CCR9, CXCR3, CXCR6, CXCR7, and XCR1 were more in gastric diffuse large B cell lymphoma with histologic evidence of MALT (DLBCL[MALT]) [54]. Notably, CXCR4 was frequently detected in nodal MALT lymphoma, and nodal DLBCL patients with bone marrow infiltration, but was not expressed in tumor cells of patients with gastric lymphoma [54]. However, Stollberg et al. revealed that CXCR4 expression was detected in 92% of the 55 cases of gastric and extragastric MALT lymphoma, and the expression of CXCR4 correlated with the proliferation index of Ki-67 [55]. In addition, the expressions of somatostatin receptors 3, 4, and 5 were frequently observed in the gastric MALT lymphoma than in the extragastric MALT lymphoma [55]. In a murine model, Wang et al. found that the use of CXCR7 inhibitors resulted in the destruction of the architecture, and a reduction in the numbers of splenic marginal zone B cells [56]. Considering that CXCL12 is the ligand that interacts with CXCR4 and CXCR7, these findings suggested that during HP infection, CXCL12 regulates the homing B cell lymphocytes to gastric mucosa, and further promotes the abnormal growth of B lymphocytes through CXCL12/CXCR4 or CXCL12/CXCR7 signaling.

References

6.        Jaffe, ES.; Harris, N.L.; Stein, H.; Isaacson, P.G. Classification of lymphoid neoplasms: the microscope as a tool for disease discovery. Blood 2008, 112, 4384-4399.

7.        Sagaert, X.; Tousseyn, T.; Yantiss, R.K. Gastrointestinal B-cell lymphomas: From understanding B-cell physiology to classification and molecular pathology. World J. Gastrointest. Oncol2012, 4, 238-249.

8.        Shaffer, A.L.; Rosenwald, A.; Staudt, L.M. Lymphoid malignancies: the dark side of B-cell differentiation. Nat. Rev. Immunol. 2002, 2, 920-932.

9.        Sagaert, X.; Sprangers, B.; De Wolf-Peeters, C. The dynamics of the B follicle: understanding the normal counterpart of B-cell-derived malignancies. Leukemia 2007, 21, 1378-1386.

10.     Bautista-Quach, M.A.; Ake, C.D.; Chen, M.; Wang, J. Gastrointestinal lymphomas: Morphology, immunophenotype and molecular features. J. Gastrointest. Oncol. 2012, 3, 209-225.

11.     Jaso, J.; Chen, L.; Li, S.; Lin, P.; Chen, W.; Miranda, R.N.; Konoplev, S.; Medeiros LJ.; Yin, C.C. CD5-positive mucosa-associated lymphoid tissue (MALT) lymphoma: a clinicopathologic study of 14 cases. Hum. Pathol. 2012, 43, 1436-1443. 

54.     Deutsch, A.J.; Steinbauer, E.; Hofmann, N.A.; Strunk, D.; Gerlza, T.; Beham-Schmid, C.; Schaider, H.; Neumeister, P. Chemokine receptors in gastric MALT lymphoma: loss of CXCR4 and upregulation of CXCR7 is associated with progression to diffuse large B-cell lymphoma. Mod. Pathol. 2013, 26, 182-194.

55.     Stollberg, S.; Kämmerer, D.; Neubauer, E.; Schulz, S.; Simonitsch-Klupp, I.; Kiesewetter, B.; Raderer, M.; Lupp, A. Differential somatostatin and CXCR4 chemokine receptor expression in MALT-type lymphoma of gastric and extragastric origin. J. Cancer Res. Clin. Oncol. 2016,142, 2239-4227.

56.     Wang, H.; Beaty, N.; Chen, S.; Qi, C.F.; Masiuk, M.; Shin, D.M.; Morse, H.C 3rd. The CXCR7 chemokine receptor promotes B-cell retention in the splenic marginal zone and serves as a sink for CXCL12. Blood 2012, 119, 465-468. 

2.       Can the authors cite references and/ or comment on, in addition to NK cells the potential involvement of tissue associated innate lymphoid cells which could, for example, act as sources of IL-22 and IFN-g (eg. MAIT cells, ILC1-3) in either promoting or restricting tumor growth.

Authors’ response:

We appreciate this pertinent suggestion. We have provided the potential mechanisms of innate lymphoid cells that are associated with the lymphomagenesis of gastric MALT lymphoma (Page 9, Lines 381-385; Page 9, Lines 388-410).

The CD56(+) natural killer (NK) cell, an important component of the innate lymphoid cells (ILCs), has the ability to modulate both humoral- and cell-mediated immune responses [97,98]. Helicobacter pylori infection enhances the production of Th2 cytokines (IL-10) that can promote the proliferation of CD56 in a gastric microenvironment [99]. Infiltration of NK cells is also detected in gastric MALT lymphoma [99].

In addition to NK cells, ILCs, lacking immune cell lineage, are divided into: (1) group 1 ILCs (ILC1s), produce IFN-γ, and require Tβ help; (2) group 2 ILCs (ILC2s), produce type 2 cytokines like IL-5 and IL-13, and require GATA3 help; and (3) group 3 ILCs (ILC3s), produce IL-17 and/or IL-22, and require RORγt help, based on the distinct production of cytokines and transcriptional factors-dependent help [98,101,102]. Previous studies suggested that persistent HP infection can allow recruitment of the ILCs to gastric mucosa, which usually lack lymphoid tissue, and promote the development of gut-associated lymphoid tissue [103,104]. Several studies with the H. species-infected animal models have demonstrated that IFN-γ can induce the formation of MALT, a precursor lesion of gastric MALT lymphoma [105,106]. However, IFN-γ also exhibits antitumor effects by producing NK cells, differentiating CD4 cells to Th1 cells, and activating cytotoxicity of CD8+ cells [107,108]. Mucosa-associated invariant T (MAIT) cells, a type of antimicrobial innate T cells, are expressed in a variety of organs, including blood, liver, lung, intestine, and stomach [109]. In the gastric mucosa of both patients and mice, D’Souza et al. showed that HP infection enhanced the production of MAIT cells [110]. Booth et al. found that MAIT cells were frequently lower in the blood of HP-positive gastritis patients than in the blood of HP-negative patients, whereas the expression of MAIT cells in gastric tissue was not different between HP-positive and HP-negative patients [111]. Using an in vitro cell study, they revealed that MAIT triggered by HP-infected macrophage harbored abilities of cytotoxicity through the production of IFN-γ, TNF-α, and IL-17 [111]. These findings indicate that ILCs are not only involved in the early process of lymphomagenesis of gastric MALT lymphoma but also limit the autonomous growth and progression of the lymphoma cells through the production of IFN-γ, NK cells, and IL-22 and interaction with adaptive immune cells. These hypotheses are supported by our findings that IL-22 expression significantly correlated with the HP-dependent status of gastric MALT lymphoma [81].

References:

98. Hazenberg, M.D.; Spits. H. Human innate lymphoid cells. Blood 2014, 124, 700-709.

101.Sonnenberg, G.F. Regulation of intestinal health and disease by innate lymphoid cells. Int. Immunol. 2014, 26, 501-507.

102.  Vivier, E.; Artis, D.; Colonna, M.; Diefenbach, A.; Di Santo, J.P.; Eberl, G.; Koyasu, S.; Locksley, R.M.; McKenzie, A.N.J.; Mebius, R.E.; et al. H. Innate Lymphoid Cells: 10 Years On. Cell 2018, 174, 1054-1066.

103.  Mebius, R. E. Organogenesis of lymphoid tissues. Nat. Rev. Immunol. 2003, 3, 292.

104.  Pearson C.; Uhlig, H.H.; Powrie. F. Lymphoid microenvironments and innate lymphoid cells in the gut. Trends. Immunol. 2012, 33, 289-296

105.  Yang L.; Yamamoto, K.; Nishiumi, S.; Nakamura, M.; Matsui, H.; Takahashi, S.; Dohi, T.; Okada, T.; Kakimoto, K.; Hoshi, N.; et al. Interferon-γ-producing B cells induce the formation of gastric lymphoid follicles after Helicobacter suis infection. Mucosal Immunol. 2015, 8, 279-295.

106.  Chonwerawong, M.; Avé, P.; Huerre, M.; Ferrero, R.L. Interferon-γ promotes gastric lymphoid follicle formation but not gastritis in Helicobacter-infected BALB/c mice. Gut Pathog. 2016, 8, 61.

107.  Ikeda, H.; Old, L.J.; Schreiber, R.D. The roles of IFN gamma in protection against tumor development and cancer immunoediting. Cytokine Growth Factor Rev. 2002, 13, 95–109.

108.  Tian, Z.; van Velkinburgh, J.C.; Wu, Y.; Ni, B. Innate lymphoid cells involve in tumorigenesis. Int. J. Cancer 2016, 138, 22-29.

109.  Le Bourhis, L.; Guerri, L.; Dusseaux, M.; Martin, E.; Soudais, C.; Lantz, O. Mucosal-associated invariant T cells: unconventional development and function. Trends. Immunol.  2011, 32, 212–218.

110.  D’Souza, C.; Pediongco, T.; Wang, H.; Scheerlinck, J.Y.; Kostenko, L.; Esterbauer, R.; Stent, A.W.; Eckle, S.B.G.; Meehan, B.S.; Strugnell, R.A.; et al. Mucosal-associated invariant t cells augment immunopathology and gastritis in chronic Helicobacter pylori infection. J. Immunol. 2018, 200, 1901‐1916.

111.  Booth, J.S.; Salerno-Goncalves, R.; Blanchard, T.G.; Patil, S.A.; Kader, H.A.; Safta, A.M.; Morningstar, L.M.; Czinn, S.J.; Greenwald, B.D.; Sztein, M.B. Mucosal-Associated Invariant T Cells in the Human Gastric Mucosa and Blood: Role in Helicobacter pylori Infection. Front Immunol. 2015, 6, 466. 
